# Spatio-temporal analysis of the distribution and co-circulation of dengue, chikungunya, and Zika in Medellín, Colombia, from 2013 to 2021

Jorge E. Salazar Flórez[ID][1]*, Berta N. Restrepo[2], Laís Picinini Freitas[3,4], Mabel Carabali[5], Gloria I. Jaramillo Ramírez[6], Cesar Garcia Balaguera[6], Brayan S. Avila Monsalve[6], Kate Zinszer[3,4]

**1** Infectious and Chronic Diseases Study Group (GEINCRO), San Martin University Foundation, Sabaneta, Colombia, **2** Instituto Colombiano de Medicina Tropical, Universidad CES, Medellín, Colombia, **3** Université de Montréal, École de Santé Publique, Montreal, Canada, **4** Centre de Recherche en Santé Publique, Montreal, Canada, **5** Department of Epidemiology, Biostatistics and Occupational Health, McGill University, Montreal, Canada, **6** Universidad Cooperativa de Colombia, Faculty of Medicine, Villavicencio, Colombia

* jorge.salazarf@sanmartin.edu.co

## Abstract

### Background

Dengue, chikungunya, and Zika present significant public health challenges in Colombia. Spatial studies help clarify the distribution and progression of these diseases over time and location. Objective to describe the spatio-temporal distribution and clustering patterns of dengue, chikungunya, and Zika in Medellín, Colombia, between 2013 and 2021, with the aim of providing baseline spatial intelligence to support future epidemiological and policy-oriented analyses.

### Methods

We analyzed dengue, chikungunya, and Zika cases in Medellín from 2013 to 2021, using weekly data from 27,459 geocoded cases across 265 neighborhoods. Cases were geocoded by neighborhood based on residential addresses in the national surveillance system (SIVIGILA). Spatio-temporal analysis identified high-risk clusters and examined the co-circulation of the diseases through multivariate analysis. We used scan statistics with a discrete Poisson model to detect high-risk clusters.

### Results

From 2013 to 2021, 26,350 dengue cases probable and confirmed were reported, with an annual incidence of 137.3 per 100,000 residents. Chikungunya and Zika emerged in 2014 and 2015, with 574 and 515 cases reported, resulting in incidences of 5.1 and 3.8 per 100,000 residents, respectively. We identified five dengue clusters and four clusters each for Zika and chikungunya, mainly in Medellín's northeast.

**Data availability statement:** The study is based on publicly accessible secondary data sources. Case data for dengue, chikungunya, and Zika can be obtained from the SIVIGILA platform (https://portalsivigila.ins.gov.co/), while population statistics are available through the Colombian National Administrative Department of Statistics (https://www.dane.gov.co/). The processed datasets used in the analyses are openly available at https://github.com/jemilios/Spatiotemporal.

**Funding:** This work was supported by a grant from the Canadian Institutes of Health Research (https://cihr-irsc.gc.ca/, grant number 428107) to K.Z. The funders had no role in study design, data collection and analysis, decision to publish, or preparation of the manuscript.

**Competing interests:** The authors have declared that no competing interests exist.

Multivariate analysis revealed six clusters, with four exhibiting high risk for all three diseases. Co-circulation of dengue, chikungunya, and Zika occurred between September 2015 and February 2017. Dengue clusters peaked between 2015 and 2016, while chikungunya and Zika peaks occurred in 2015 and 2016, respectively.

## Conclusions

This study advances understanding of spatio-temporal dynamics in arbovirus transmission in Medellín, highlighting high-risk clusters for dengue, chikungunya, and Zika and their collective presence. Our findings support targeted public health interventions to mitigate these diseases.

## Author summary

This study presents a comprehensive spatio-temporal analysis of the distribution and co-circulation of dengue, chikungunya, and Zika in Medellín, Colombia, from 2013 to 2021. By examining over 27,000 geocoded cases across 265 neighborhoods, it provides crucial insights into the spatial dynamics of these arboviruses. The research identifies high-risk clusters for each disease and highlights zones with concurrent transmission of all three viruses, particularly concentrated in Medellín's northeastern region. These findings underscore the importance of simultaneous surveillance for multiple vector-borne diseases, providing valuable data to inform targeted public health interventions. This analysis addresses a critical gap in understanding the intra-urban spread of arboviruses in tropical cities. By mapping high-risk clusters, the study offers actionable insights for health authorities to develop more effective, location-specific disease control strategies. This work is essential for advancing knowledge on arbovirus transmission and enhancing urban preparedness against multiple vector-borne threats.

## Introduction

Recently, vector-borne diseases, especially dengue, chikungunya, and Zika, have become a significant global public health concern [1]. Dengue alone accounts for approximately 390 million new cases worldwide each year [2]. However, about 75% of these cases remain asymptomatic and are not reported in the official surveillance records [2]. A recent global meta-analysis estimated that approximately 59% (95% CI: 44–75%) of dengue infections are asymptomatic, with higher proportions during outbreak periods [3,4]. For chikungunya, evidence from systematic reviews and serosurveys indicates that about 25%–40% of infections show no clinically recognizable symptoms [5]. Zika virus infections are also largely silent, with pooled estimates indicating around 59% of infections are asymptomatic, though individual studies range from 33% to 88% [6]. Nearly 96 million infections present symptoms annually,

requiring health system intervention [7]. Latin America is one of the hardest-hit regions, representing 14% of all global cases [8].

Zika and chikungunya are less prevalent but have led to significant epidemics, causing lasting impacts in affected regions. Zika virus has been linked to various neurological complications, including microcephaly in children born to individuals infected during pregnancy [9]. There have been reports of locally transmitted cases in 89 countries since its discovery in 1952 [10], with epidemic outbreaks notably in 2013 and 2015 in the Western Pacific [11] and the Americas [12,13], respectively, reporting up to 30,000 cases annually. Currently, 61 countries have reported no cases of the disease despite the continued presence of the *Aedes aegypti* vector [10].

Chikungunya, similar to Zika, has expanded vastly in recent decades, with a significant increase in Asia, Africa, and America [14–16]. Chikungunya was identified in Cambodia in 1961. Initial outbreaks were recorded in Kenya and the Comoros Islands in 2004 for Africa [15]. It resurfaced in Sri Lanka in 2007 with over 37,000 suspected cases [14]. Europe recorded 205 cases in Italy in 2007 [17]. By 2010, France, too, had local transmission cases [18]. In December 2013, the first outbreaks in the Americas were identified, and by December 2015, nearly one million cases were reported [19].

In Latin America, these arboviruses have had a critical epidemiological impact. Chikungunya, introduced in 2013, peaked in 2015, accounting for nearly one million cases and 7,126 confirmed deaths [20,21]. For Zika, after the first case in the Americas was reported in Chile in March 2014, the virus spread swiftly across the continent, severely affecting Brazil and Colombia, with the highest incidence peaks occurring in 2015 and 2016 [22]. For instance, dengue peaked in 2019 with over 3.1 million cases and approximately 1,534 confirmed deaths [23].

In Colombia, these diseases continue to pose a significant public health challenge. The country ranks as the second-highest in terms of Zika cases in Latin America [24], and dengue is endemic throughout the nation, with an average of over 50,000 reported cases annually [25]. Both Chikungunya and Zika primarily spread from regions along the Atlantic coast and southwest to other parts of the country [26,27]. The arbovirus situation highlights the need to understand their spatial patterns, which will help us better understand their distribution [26,28].

Medellín is Colombia's second-largest city and one of the most urbanized and densely populated municipalities in the country, with over 2.5 million inhabitants concentrated in 376 km². Its tropical sub-humid climate, altitude between 1,500 and 1,800 meters, and persistent presence of *Aedes aegypti* mosquitoes create optimal ecological conditions for arbovirus transmission, including dengue, chikungunya, and Zika [29–31]. Previous studies conducted in Colombian cities have shown that arboviruses such as dengue, chikungunya, and Zika exhibit heterogeneous patterns of transmission at the intra-urban level. In Medellín, the distribution of cases has historically concentrated in neighborhoods with high population density, poor housing conditions, and limited access to public services.

In recent decades, Medellín has experienced cyclical outbreaks of dengue and was one of the earliest urban epicenters for chikungunya (2014) and Zika (2015) introduction in Colombia [25–27,32]. Furthermore, the city's marked socioeconomic heterogeneity across comunas and neighborhoods allows for the study of health inequities in disease distribution, as lower-income areas often face higher entomological risk due to poor water storage and limited vector control coverage [33,34]. Conducting a fine-scale spatio-temporal analysis in Medellín is relevant not only to guide local interventions but also because the city serves as a model for arboviral dynamics in rapidly urbanizing tropical settings.

Understanding the spatio-temporal dynamics of these arboviruses is paramount for discerning propagation patterns and shaping effective control and prevention strategies [35,36]. Arboviruses are not uniformly distributed in time or space; certain geographical areas may be more susceptible due to environmental factors—such as temperature, rainfall, and urban infrastructure—that favor the proliferation of vector mosquitoes, as well as societal factors like population density and health practices [37–39]. Moreover, distinct temporal transmission patterns exist, with certain times of the year or cycles over multiple years indicating higher disease incidences. Acknowledging these transmission hotspots could enhance future outbreak predictions, enabling targeted resource allocation for prevention and control efforts [26,40].

This study aims to provide a detailed description of the spatio-temporal clustering of dengue, chikungunya, and Zika in Medellín from 2013 to 2021. Rather than seeking generalizable associations or predictive patterns, our objective is to generate fine-scale evidence to support local decision-making and to inform future analytic studies exploring the structural and ecological determinants of arboviral transmission in urban settings.

## Methods

### Ethics statement

The research was approved by the ethics committee of the Colombian Institute of Tropical Medicine (Instituto Colombiano de Medicina Tropical, ICMT), along with an agreement of confidentiality to handle sensitive data (home addresses) with Medellin's local health secretariat as documented in the minutes from May 25, 2021. Informed consent was not required as the study used secondary data. ChatGPT was used to correct and adjust the style of the English language.

We conducted an ecological study of the spatio-temporal distribution of reported dengue, chikungunya, and Zika cases in Medellín, Colombia, from 2013 to 2021. The analysis used aggregated data by neighborhood of residence and epidemiological week of symptom onset, without linking individual-level exposures or outcomes. The analysis included both probable and confirmed cases reported to the national surveillance system SIVIGILA (*Sistema Nacional de Vigilancia en Salud Pública*,https://portalsivigila.ins.gov.co) and acquired from the Medellín Health Secretariat.

### Area and study population

Medellín is the capital of the Department of Antioquia and Colombia's second-largest city. It spans an area of 376.2 km$^2$ and houses a population exceeding 2.5 million individuals, based on the projected numbers for 2023 from the National Administrative Department of Statistics (*Departamento Administrativo Nacional de Estadística*, DANE) [29,30]. The city's administration divides Medellín's territory into 21 units known as 'Comunas'. A *comuna* is equivalent to a district and encompasses several neighborhoods (*barrios*). In this study, spatial clusters were identified at the neighborhood level, while the spatial descriptions refer to the corresponding *comunas* (districts) in which those neighborhoods are located. Medellín's altitude ranges between 1,500 and 1,800 meters above sea level, at geographic coordinates 75º 34' 05" W (longitude) and 6º 13' 55" N (latitude). Medellin experiences a sub-humid subtropical climate, with urban temperature fluctuations between 16 and 28 °C. This study focused on 265 neighborhoods within 16 urban Comunas, which covered an area of 107 km$^2$ [31]. The city was labeled as dengue hyperendemic by the country surveillance system as of 2013. Whereas chikungunya was first detected in Medellín in August 2014 and local transmission of Zika was confirmed in the city in September 2015 (SIVIGILA, https://portalsivigila.ins.gov.co/).

### Data analysis

Cases were obtained from SIVIGILA, Colombia's national surveillance system, which includes both probable and confirmed cases of dengue, chikungunya, and Zika. Probable cases are defined by clinical criteria established by the Ministry of Health, and confirmed cases require laboratory tests such as ELISA, RT-PCR, or viral isolation. Definitions vary by virus but commonly include fever and specific symptom combinations (e.g., rash, arthralgia, or conjunctivitis), with confirmation based on epidemiological links or positive laboratory results. Full case definitions and diagnostic criteria are provided in national protocols [25,27,32].

Cases were geocoded by neighborhood based on the individuals' usual place of residence, as recorded in the SIVIGILA database. We employed MAPGIS software version 5.0 (https://www.medellin.gov.co/mapgis9/mapa.jsp?aplicacion=41), which utilizes cadastral sources, power facilities, road networks, proximate addresses, and city toponymy for geocoding. The analysis considered only validated cases with addresses within urban zones and excluded duplicate records. Of the 34,086 records representing dengue, chikungunya, and Zika cases reported to SIVIGILA by the Medellín

City Health Department, 27,439 cases (80.4%) were successfully geocoded to 265 urban neighborhoods. The map was generated using municipality boundary basemaps from GADM (www.gadm.org) and open data from the Medellín city portal (https://www.medellin.gov.co/geomedellin/datosAbiertos/1043).

A preliminary examination of the distribution of cases for the three diseases was conducted. To achieve this, cases were standardized. The transformation was performed for visualization purposes, to facilitate the comparison of temporal trends across arboviruses on a standardized scale. The Average Annual Incidence for each disease was calculated using the DANE's population projections [29] and the cases reported by SIVIGILA. Following this, we investigated the spatio-temporal distribution of each arbovirus using Kulldorff's scan statistics, implemented in SaTScan software (version 10.1) [41]. Specifically, we applied: (1) a univariate spatio-temporal scan to identify clusters for each disease independently, and (2) a multivariate spatio-temporal scan statistic to detect clusters with simultaneous co-circulation of dengue, chikungunya, and Zika. Both analyses were based on a discrete Poisson model, adjusting for the at-risk population (neighborhood population) and identifying high-risk areas where the observed number of cases significantly exceeded the expected counts [42–44].

Kulldorff's scan statistics were utilized to detect space-time clusters for each disease in separate and simultaneously using multivariate analysis. These methods assume that the number of observed cases in each area follows a discrete Poisson distribution, where the expected number of cases is proportional to the population at risk. This model is Bayesian hierarchical. The analysis compares the risk inside a scanning window (in space and time) to the risk outside, identifying clusters where the observed count significantly exceeds the expected count under the null hypothesis of equal risk. Clusters of varying sizes are detected up to a predefined maximum. This analysis can identify clusters of any size up to a specified maximum size. Clusters are detected by comparing the risk inside the cylinder to those outside it. In this context, the "cylinder" represents the scanning window used by Kulldorff's method, with a circular geographic base (space) and a vertical dimension corresponding to time. Relative risks were presented with confidence intervals. The 95% confidence intervals for relative risks (RR) were calculated using the standard error of the logarithm of RR, with equation 1:

$$SE\,(\ln RR) = \sqrt{\frac{O}{E} + \frac{1}{E}},$$

(1)

$O$ is the number of observed cases, and $E$ is the number of expected cases in the cluster. The confidence interval limits were obtained as:

$$= (\ln RR) \pm Z * SE\,((\ln(RR)),$$

(2)

with $Z = 1.96$ for a 95% confidence level. The final RR confidence limits were calculated by exponentiating the limits of the logarithm of RR:

$$Lower\ limit = e^{((\ln RR) - Z*SE\,((\ln(RR)))}, \text{ and } Upper\ limit = e^{((\ln RR) + Z*SE\,((\ln(RR)))}$$

(3)

Clusters were ordered according to the likelihood ratio, with the ones with the highest maximum likelihoods considered the most probable clusters [42,43]. The likelihood function is proportional to

$$\left(\frac{c}{E[c]}\right)^c \left(\frac{C-c}{C-E[c]}\right)^{C-c} I\,(c > E[c])\,,$$

(4)

$C$ represents the total number of cases, $c$ is the observed number within the window, and $E[c]$ is the expected number under the null hypothesis. The null hypothesis was that virus risk remains the same inside and outside the scanning

 **Neglected Tropical Diseases**

window in space; since the analysis focuses on the total observed cases, $C - E[c]$ represents the expected number of cases outside the window. $I(c > E[c])$ is an indicator function. The expected number of cases in each area under the null hypothesis is calculated using the equation:

$$E[c] = p * \frac{C}{P}$$

(5)

Where $c$ denotes the observed number of cases, and $p$ represents the population in each district (comuna), $C$ and $P$ refer to the total number of cases and total population, respectively.

The spatial exploratory window was set to the centroid of each location, with a pre-set maximum size of a 1 km radius and 25% of the total population at risk. Some neighborhoods in the city, particularly those in marginalized areas, have a higher population density, and approximately 56% of the population resides in seven Comunas of the city (comunas 3, 6, 7, 8, 9,13, and 16) [31]. In effect, focusing solely on the maximum population at risk would lead to clusters of highly varied sizes. Therefore, we considered a cylinder radius of one kilometer. The temporal exploratory window spanned from 2 to 100 weeks for dengue, 2–13 weeks for chikungunya, and 2–52 weeks for Zika, and a minimum requirement of 900, 12, and 30 cumulative cases for dengue, chikungunya, and Zika, respectively (Table 1). Estimations were performed using multiple parameters until the model with the greatest interpretability was identified. Each model underwent 999 Monte Carlo simulations to assess the statistical significance of the identified clusters, with only those exhibiting a p-value below 0.05 being reported.

Cluster detection and evaluation were conducted using SaTScan software, version 10.1 [45]. These clusters were visualized on corresponding maps using Python 3 with the Matplotlib library [46].

## Results

### Average annual incidence

Between January 2013 and December 2021, a total of 26,350 dengue cases were reported across 265 neighborhoods in Medellín, Colombia. With an average population of 2,131,159 urban residents during this period, the average annual incidence rate of dengue was 137.3 cases per 100,000 residents. For chikungunya, 574 cases were reported between September 26, 2014, and December 31, 2019, leading to an average annual incidence rate of 5.1 cases per 100,000 residents. Similarly, from September 17, 2015, to December 31, 2021, 515 cases of Zika were reported in the same areas, corresponding to an average annual incidence of 3.8 cases per 100,000 residents.

### Time series

Fig 1 illustrates the standardized series of dengue, chikungunya, and Zika cases during our research period. Each of the three diseases demonstrated substantial fluctuations in cases over time. Two important chikungunya increases were observed in 2015 and 2016, while for Zika only one epidemic wave was observed, in 2016. 2016 also marked the year with the most pronounced cases peaks for dengue, while chikungunya peaked in 2015. Following these peaks, the cases

**Table 1. Spatio-temporal scan parameters used for each arbovirus in SaTScan analysis.**

| Virus | Min. cases | Temporal windows | Spatial Size |
|---|---|---|---|
| Dengue | 900 | 2-100 weeks | 1 km/ 25% of the population |
| Chikungunya | 12 | 2-13 weeks | 1 km/ 25% of the population |
| Zika | 30 | 2-52 weeks | 1 km/ 25% of the population |

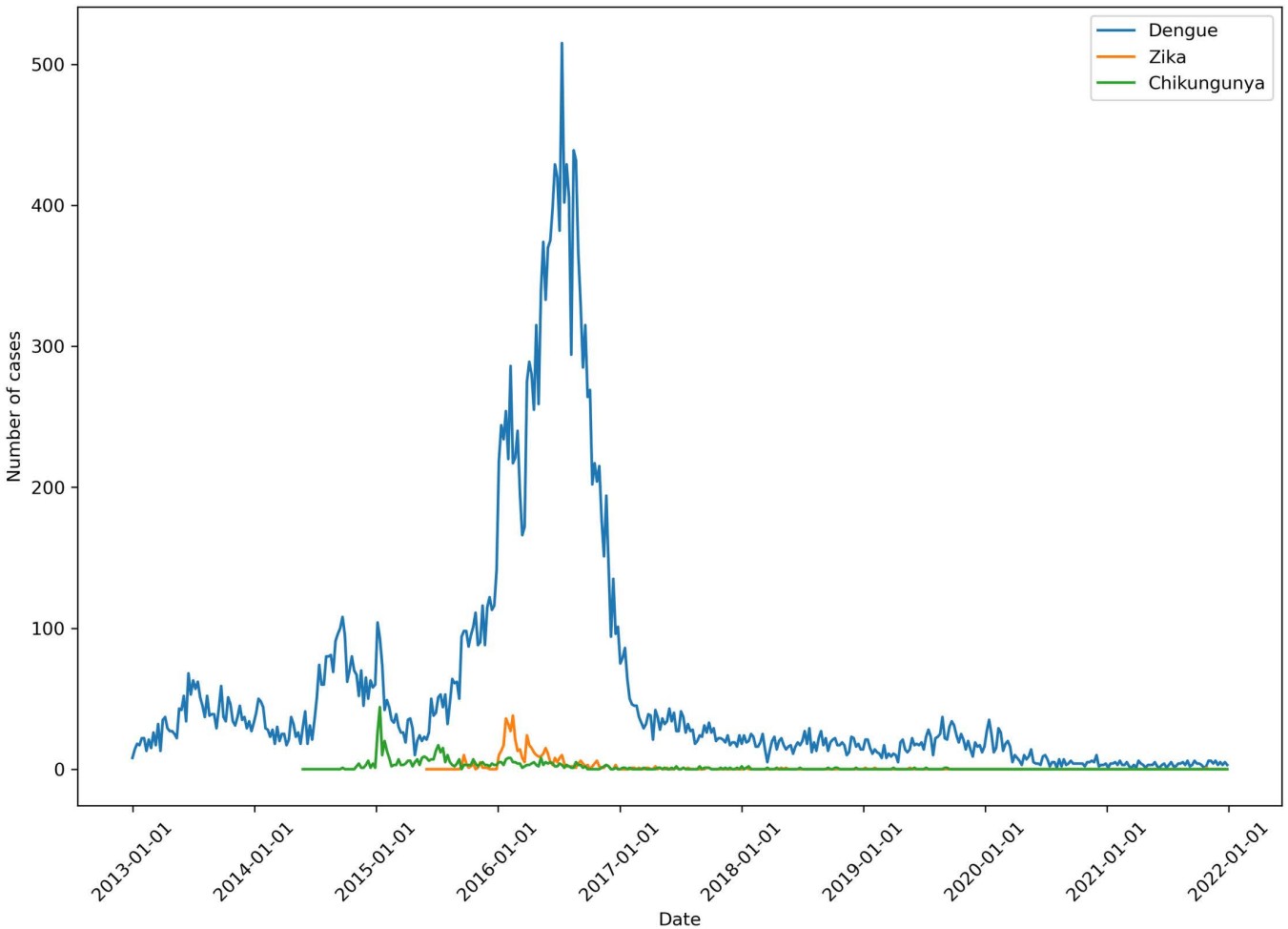

**Fig 1. Standardized time series showing weekly reported cases of dengue, chikungunya, and Zika in Medellín, Colombia, from 2013 to 2021.**

of all three diseases declined, with very few cases of chikungunya and Zika being reported, while dengue cases remained being detected at a roughly constant rate, without apparent peaks patterns.

## Cluster identification

We identified five spatio-temporal clusters for dengue and four clusters for chikungunya and Zika, each. Table 2 provides a median and range-based summary of the clusters' characteristics. Between 2013 and 2021, five significant spatiotemporal clusters of dengue were identified in Medellín, each with a distinct temporal distribution during the largest recorded epidemic in 2016. The first cluster occurred during the initial rise in cases, while the second coincided with the epidemic peak. The third and fourth clusters reflected transitional and declining phases, respectively, and the fifth emerged during the final epidemic period (S1 Fig). For Zika virus, which circulated predominantly between 2015 and 2016, four temporal clusters were detected, encompassing the rise, peak, and subsequent decline of the outbreak (S2 Fig). In contrast, Chikungunya clusters were mainly concentrated in 2015, the year with the highest disease burden, indicating an acute emergence and rapid decline. These temporal patterns illustrate the distinct epidemic dynamics of each arbovirus and suggest a staggered, yet partially overlapping, circulation within the city (S3 Fig).

**Table 2. Cluster median and range for dengue, chikungunya, and Zika in Medellín, Colombia, from 2013 to 2021.**

|  | Dengue | Chikungunya | Zika |
|---|---|---|---|
| Clusters | 5 | 4 | 4 |
| Neighborhoods | 10 (8 - 14) | 10 (6 - 12) | 11 (8 - 12) |
| Duration in Weeks | 57 (48 - 62) | 12 (3 - 15) | 34 (20 - 48) |
| Relative Risk | 7.89 (4.64 - 14.36) | 17.01 (10.93 - 34.72) | 16.50 (7.29 - 34.82) |
| Population | 115,550 (70,871 -169,197) | 111,874 (60,608 -122,415) | 97,525 (70,871 – 122,701) |
| Observed Cases | 1235 (903 - 1448) | 14 (12 - 16) | 32 (30 - 35) |

NOTE: Median (Range [minimum - maximum]).

## Dengue spatial and temporal distribution

Five clusters were found for the entire analysis period. All clusters emerged between 2015 and 2016. Cluster 1 encompassed eight neighborhoods spanning Villa Hermosa to Buenos Aires Comunas, registering a relative risk (RR) of 14.36 (95% CI: 11.79; 17.49) over 57 weeks between September 2015 and October 2016. Cluster 2, formed by ten neighborhoods, was situated in the city's central region between Buenos Aires and La Candelaria Comunas. The RR was estimated at 9.51 (95% CI: 8.01; 11.30) and the cluster persisted for 59 weeks from September 2015 through October 2016. Cluster 3 was positioned between the Manrique, Aranjuez, and Villa Hermosa Comunas, incorporating eleven neighborhoods, mostly from Manrique's district. The RR was estimated at 7.89 (CI 95%: 6.65, 9.36) and spanned 49 weeks from December 2015 to December 2016. Clusters 4 and 5 were in the city's north, in the Popular and Doce de Octubre Comunas. Cluster 4, with an RR of 4.64 (CI 95%: 4.07; 5.29), comprised 14 neighborhoods and lasted 62 weeks from October 2015 to December 2016. Conversely, Cluster 5 incorporated ten neighborhoods, exhibited an RR of 5.22 (CI 95%: 4.44; 6.13) and ran for 48 weeks from January to December 2016 (Fig 2A and 2B).

The spatiotemporal analysis of dengue cases in Medellín revealed marked changes in the geographic distribution and intensity of clusters over the studied periods pre-epidemic, during the epidemic (2016) and post-epidemic. From 2013 to 2015 (S4 Fig), five significant space-time clusters (S4A Fig) were identified, primarily in the northeastern and northwestern sectors of the city, with relative risk (RR) values exceeding 10 in the most affected neighborhoods (S4B Fig). During the subsequent period from 2013 to 2014 (S5 Fig), the number of clusters decreased to four, yet the highest RR values persisted in the central-northeast corridor, suggesting consistent transmission. Notably, in the most recent period analyzed (S6 Fig), five clusters reemerged, including high-risk zones in the eastern and southeastern communes, accompanied by an increase in relative risk magnitude. These patterns underscore the persistence and spatial migration of high-transmission zones, likely influenced by changes in urban dynamics, population mobility, and vector control coverage.

## Chikungunya spatial and temporal distribution

Four chikungunya clusters were observed, but these were identified in 2015. All four clusters were in the north of the city. Cluster 1, with six neighborhoods in the Santa Cruz district, had an RR of 21.57 (CI 95%: 1.99; 233.18) and lasted 12 weeks between June and August. Cluster 2 covered nine neighborhoods, eight of which were in the Robledo district and one in the Doce de Octubre Comunas; it had the highest RR of 34.72 (CI 95%: 1.21, 1000.53) and the shortest duration, lasting three weeks in January. Cluster 3, with 12 neighborhoods in Aranjuez and one in Manrique, had an RR of 12.44 (CI 95%: 2.11; 73.39) and lasted 15 weeks from May to August. Finally, Cluster 4, mainly in the Castilla Comunas, which extends to Aranjuez and Santa Cruz, included ten neighborhoods, had an RR of 10.93 (CI 95%: 1.58; 75.80), and lasted 12 weeks from May to July (Fig 2C and 2D).

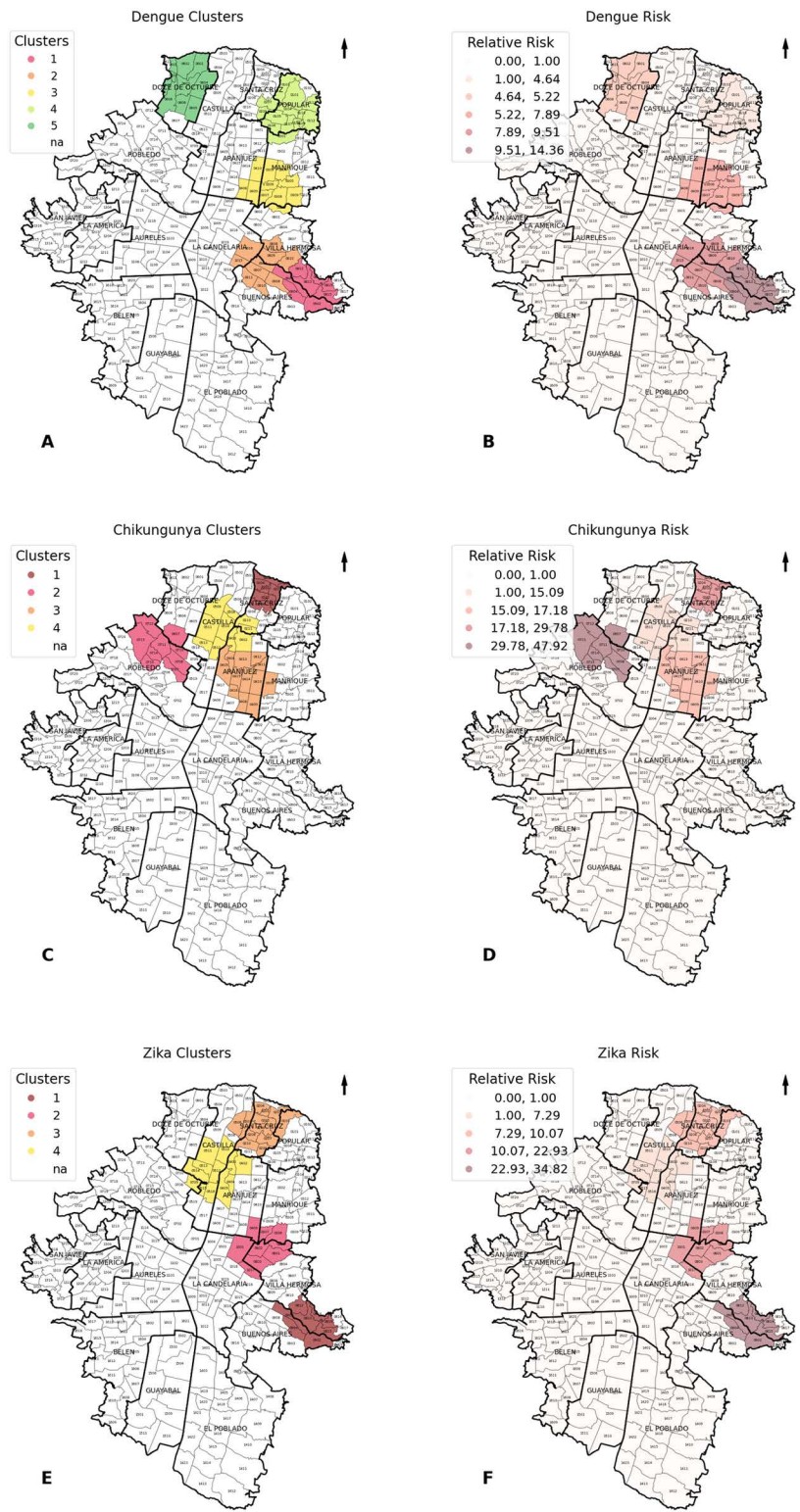

**Fig 2. Space-time clusters and relative risk of dengue, chikungunya, and Zika clusters in Medellín, Colombia, between 2013 and 2021.** Notes: Panel A and B: Space-time clusters (A) and relative risk (B) of dengue clusters. Panel C and D: Space-time clusters (C) and relative risk (D) of chikungunya clusters. Panel E and F: Space-time clusters (E) and relative risk (F) of Zika clusters. The map was generated using municipality boundary basemaps from GADM (www.gadm.org) and open data from the Medellín city portal (https://www.medellin.gov.co/geomedellin/datosAbiertos/1043).

## Zika spatial and temporal distribution

Four Zika clusters were detected in 2016, reflecting dengue's pattern. Cluster 1 included eight neighborhoods between the Villa Hermosa and Buenos Aires Comunas, with an RR of 34.82 (CI 95%: 4.95; 245.01) over 20 weeks between January and June. Cluster 2 covered nine neighborhoods between Villa Hermosa and La Candelaria Comunas, with an RR of 22.93 (CI 95%: 4.19, 125.58) over 26 weeks between January and July. Cluster 3 was located between the Santa Cruz and Doce de Octubre Comunas in the north of the city, including 12 neighborhoods and an RR of 10.07 (CI 95%: 3.45; 29.40); its duration was 42 weeks between January and November. Cluster 4, in the Aranjuez district, included 12 neighborhoods, had an RR of 7.29 (CI 95%: 2.66; 19.95), and lasted 48 weeks from January to December (Fig 2E and 2F).

## Multivariate analysis (co-circulation)

The spatio-temporal multivariate assessment for Medellín identified five clusters. Four of these exhibited dengue, chikungunya, and Zika co-circulation, while Cluster 5 showed co-circulation of dengue and Zika. These clusters persisted for 52 weeks (Cluster 3) and 65 weeks (Cluster 5) between September 2015 and February 2017. Clusters 1 and 2 included eight and ten neighborhoods, respectively, between Buenos Aires and Villa Hermosa for Cluster 1 and La Candelaria and Villa Hermosa for Cluster 2. Clusters 3 and 5 each included eleven and ten neighborhoods, mainly in the Manrique and Doce de Octubre Comunas. Meanwhile, Cluster 4 had 14 neighborhoods, of which 11 were in the Popular district, two in Santa Cruz, and one in Manrique (Fig 3).

## Discussion

Our study shows that dengue, chikungunya, and Zika co-circulated between September 2015 and February 2017 in Medellín. We found 5 clusters for dengue that peaked in 2015 and 2016, with evidence cyclical virus. For chikungunya and Zika, we found 4 clusters with peak incidence in 2015 and 2016, respectively. Our results indicate that high-risk clusters were concentrated in the city's downtown, and northeastern Comunas.

Dengue cases showed no consistent peaks pattern in the visual time series, but exhibited cyclical peaks approximately every two to three years. Between 2013 and 2021, Medellín averaged 137.3 dengue cases per 100,000 inhabitants annually. Another spatio-temporal study in Medellín covering 2007–2017 reported a similar trend, characterized by cyclical dengue outbreaks every 2–3 years and concentrated incidence peaks, consistent with the temporal pattern observed in our analysis [34]. Colombia experienced significant outbreaks in 2010, 2013, and 2015, with around 150,000 cases each year [47,48], and 127,925 cases were reported in 2023, exceeding the 2015 (96.844 cases) epidemic by 32.0% [49]. Other Colombian places, such as Ibague, Cali, and Bucaramanga, also showed significant outbreaks in 2013, 2015, and early 2016 [50–53]. The cases in Medellín mirrors those observed in Colombia and other Latin American countries, such as Brazil, where cities like Rio de Janeiro reported high incidences between 2011–2013 and 2015–2016 [54,55]. In our study, this behavior was evident up to 2016, after which dengue incidence remained low and irregular.

As expected, following its introduction in 2014, Chikungunya exhibited a sharp initial epidemic in 2015, followed by a steep decline in reported cases after 2016 [26]. High Zika incidence was reported in densely populated, lower-altitude municipalities in Colombia with lower weekly rainfall compared to other endemic areas [56]. While Medellín does not have particularly low rainfall, its high population density and favorable ecological conditions for Aedes aegypti may have contributed to the observed transmission. Between 2013 and 2021, Medellín had an average annual chikungunya rate of 5.1 cases per 100,000 inhabitants [26]. A national study reported increased chikungunya and Zika incidence in Colombia's Andean region between 2014 and 2017 [57], which coincides temporally with the epidemic peaks observed in Medellín. Colombia and Venezuela reported the highest incidence rates in the Andean subregion between 2016 and 2020 [58]. Differences in incidence patterns reported across studies, time periods, and regions may be explained by diagnostic challenges, underreporting, vector adaptability, population immunity, and environmental conditions [26,34,37,55,59–61].

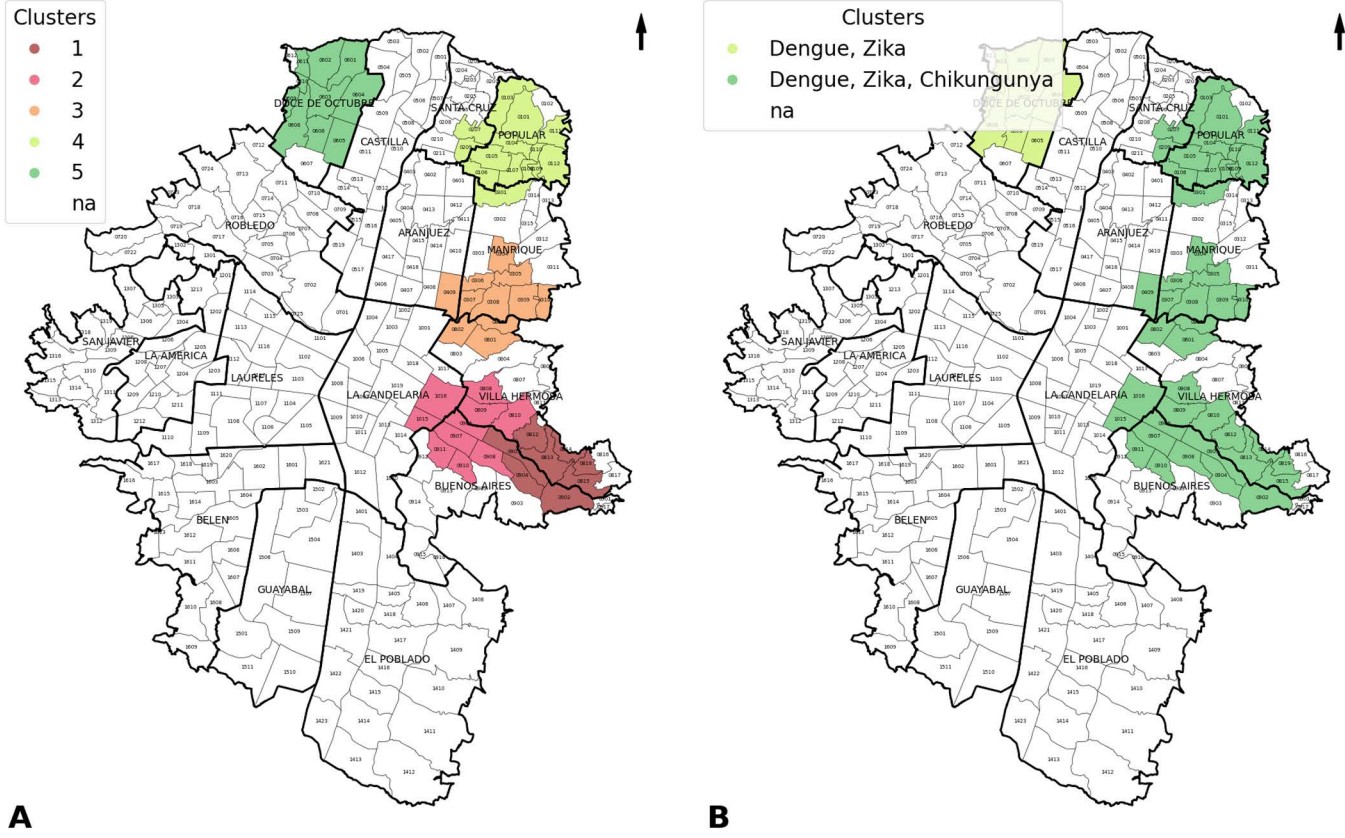

**Fig 3. Location of the multivariate spatio-temporal clusters for dengue, chikungunya, and Zika, Medellín, Colombia, 2013 to 2021.** Notes: Panel A: Space-time clusters. Panel B: Space-time clusters (co-circulation of dengue, chikungunya, and Zika). The map was generated using municipality boundary basemaps from GADM (www.gadm.org) and open data from the Medellín city portal (https://www.medellin.gov.co/geomedellin/datosAbiertos/1043).

The year 2016 represents a critical point in the analysis, as it corresponds to the highest reported incidence of dengue, chikungunya, and Zika in Medellín, marking the period of greatest co-circulation and overlapping transmission of the three arboviruses.

Vector dynamics, including their density and adaptation to urban conditions, play a key role in shaping arbovirus transmission patterns over time and space [62]. Monitoring vector behavior and assessing the risk of transmission through entomological indices are crucial in this process. In 2022, in the department of Antioquia—where Medellín is located— entomological risk levels for *Aedes aegypti* based on the household positivity index classified eight municipalities as high or medium risk, while 41 were considered low risk [63]. A study between 2013–2015 in Medellín showed that neighborhoods with a higher incidence of dengue had a significant number of households storing water (OR: 1.69; 95% CI 1.11; 2.57) and more homes with adult mosquitoes (OR: 2.13; 95% CI 1.29; 3.50) [33].

As previously mentioned, dengue incidence in Medellín has historically shown peaks every two to three years. However, a visual inspection of reported case data suggests a decline in incidence from 2017 to 2022, though no formal trend analysis was performed. Other studies have shown a reduction in the incidence of dengue in the city, particularly in 2020 and 2022 [64,65]. This decline could be partially explained by the introduction of Wolbachia into *Aedes aegypti* mosquitoes, aimed at reducing their ability to transmit dengue and other arboviruses. In Medellín, *Wolbachia*-infected mosquitoes were released progressively across different comunas between October 2016 and April 2022 [65]. Given that not all

areas were covered simultaneously, this intervention may have influenced the spatial distribution and intensity of dengue clusters, contributing to the observed decline in incidence [65]. However, the pandemic may have overshadowed cases and complicated the surveillance of other febrile illnesses [66–68]. However, in 2023 and 2024 an increased number of dengue cases were reported in the region [69].

We identified five dengue clusters that emerged between 2015 and 2016, coinciding with one of the highest peaks in dengue incidence in the time series. Although the scan statistic does not directly assess periodicity, the temporal concentration of these clusters —together with historical outbreak records and visual inspection of incidence trends— suggests a cyclical transmission pattern, with peaks occurring approximately every two to three years. On average, dengue clusters lasted 55 weeks, compared to 10.5 weeks for chikungunya and 34 weeks for Zika. These durations differ from those reported in other settings; for instance, an analysis in the department of Cauca found dengue clusters lasting one to two years [51], while a national study reported average durations of 25 weeks for dengue, 19.5 weeks for chikungunya, and 14 weeks for Zika [26]. Such variation may reflect differences in geographic scale (municipality vs. neighborhood), timing of virus introduction, population immunity, and the unique urban transmission dynamics in Medellín. These comparisons provide valuable context for understanding the intensity and persistence of arboviral clustering in large urban centers.

Our findings indicate that several high-risk clusters occurred in the northeastern comunas of Medellín, areas characterized by high population density and socioeconomic disadvantage. This pattern is consistent with national and local studies showing that arboviruses -whether circulating individually or concurrently- are more prevalent in marginalized urban settings [26]. A previous study conducted between 2008 and 2011 identified the southwestern zone of Medellín, particularly the Belén neighborhood, as a hotspot for dengue outbreaks [70]. However, more recent data suggest a spatial shift toward the northeast, where living conditions remain poor. This shift is supported by recent research highlighting a concentration of arboviral cases in the northeastern comunas [34], in line with the clusters identified in our study and reinforcing the link between socioeconomic vulnerability and arboviral burden in the city [52]. Similarly, a study in Bucaramanga identified four critical clusters, with notable differences in dengue risk between urban core and suburban areas [53]. At the national level, seven clusters with co-circulation of arboviruses were reported, along with 21, 16, and 13 individual clusters for dengue, chikungunya, and Zika, respectively [26].

The co-circulation of dengue, chikungunya, and Zika is a recognized phenomenon in Latin America [36,37,39,54,71,72]. In Medellín, multivariate clusters were concentrated in northeastern *comunas*, likely influenced by socioeconomic factors such as high population density, poor housing, irregular water access, and limited vector control. Although the three viruses share *Aedes aegypti* as a common vector, simultaneous clustering was not uniform across the city. This may reflect differences in virus introduction dengue being endemic, while chikungunya and Zika emerged in 2014 and 2015 [24] as well as local variation in immunity, vector density, mobility, and interventions like *Wolbachia* [65]. These findings suggest that arbovirus clustering is shaped by overlapping social, environmental, and biological factors, rather than occurring at random.

Socioeconomic, environmental, and biological factors appear to shape the co-circulation of arboviruses in Medellín. The viruses primarily emerged in socioeconomically disadvantaged neighborhoods, particularly in the northeastern comunas of the city. Prior studies have shown that the areas identified as high-risk clusters exhibit low living condition indices, often with composite scores below 40 on a 0–100 scale, where lower values denote poorer access to health services, education, housing, and employment opportunities [33,34]. An exception is the Buenos Aires district, where 59% of the population belongs to the upper-middle socioeconomic stratum and the living condition index approaches 50 [73]. However, the presence of arbovirus clusters in this area may be attributed to its high population density and substantial intra-urban mobility, which can facilitate viral transmission even in neighborhoods with relatively better socioeconomic indicators.

The co-circulation of dengue, chikungunya, and Zika in Medellín was predominantly concentrated in northeastern comunas, characterized by high population density, poor housing conditions, and limited public service access. In neighborhoods such as Santa Cruz and La Candelaria, where all three viruses overlapped, the affected areas ranged from

2.2 km² to 7.36 km², with population densities reaching up to 447,542.6 inhabitants in Popular [73]. These conditions likely facilitate viral transmission and complicate vector control efforts and healthcare access. Similar patterns have been observed in other Colombian cities —such as Cali— where dengue outbreaks have been linked to socioeconomic and environmental factors like limited piped water, poor waste disposal, and overcrowding [74], and in Bucaramanga, where cluster risk varied across urban and suburban zones [50,51]. Comparable evidence from Fortaleza (Brazil) and Mexico confirms that arboviral transmission disproportionately affects low-income populations with limited access to basic services [34,59].

Another crucial factor is climate variables, such as temperature and rainfall, which have been shown to impact the transmission of dengue, chikungunya, and Zika [37,59,71]. Although this study focuses on an intra-urban setting, and broad climatic conditions may be relatively stable, we acknowledge that Medellín's complex topography may lead to local microclimatic variations that could influence arbovirus transmission. It's essential to consider the growing adaptability of the vector to higher altitudes in the Andean region, which may have implications for transmission in mountainous areas of Medellín [58]. It was determined that over fifty years, the city of Medellín experienced climate change, with an increase in the average temperature of 0.8°C, a rise of 1.3°C in the minimum temperature, a 0.5°C increase in the maximum temperature, and a downward trend in relative humidity, with an average decrease of 2.3% [75]. It is important to note that the cited study reports climatic trends in Medellín up to 2010 [75], and conditions have likely changed in the past 15 years due to ongoing urban expansion and climate dynamics. These changes —such as rising temperatures and decreasing humidity— may enhance the adaptability of *Aedes aegypti* to higher altitudes, potentially expanding its ecological niche and contributing to arbovirus transmission in previously unaffected areas of Medellín [57].

From a public health perspective, identifying high-risk clusters of arbovirus transmission at the intra-urban level enables more efficient allocation of resources for vector control, surveillance, and community engagement [24,60,61,76]. The spatial overlap of multivariate clusters with socioeconomically disadvantaged areas suggests that interventions should be tailored to specific local contexts [28,60]. To enhance the explanatory power and actionability of these findings, future studies should incorporate additional variables such as rainfall, temperature, entomological indices, housing quality, and mobility patterns to model the drivers of clustering and guide integrated control strategies.

A key limitation of this study is the reliance on routine surveillance data, which includes both probable and confirmed cases. Given the overlapping clinical presentations of dengue, chikungunya, and Zika, particularly in the absence of laboratory confirmation, misclassification is likely and may have influenced the spatial and temporal patterns observed. Most cases were identified based on clinical suspicion, with only a small proportion confirmed by ELISA or PCR testing. Additionally, many individuals with mild or asymptomatic infections may not have sought medical care, contributing to underreporting and limiting the representativeness of the dataset, which only reflects those who accessed healthcare services [61]. These limitations are compounded by the absence of seroprevalence studies in Medellín, which restricts our ability to validate findings or estimate the true burden of infection and extent of underreporting.

Part of the data collection also coincided with the COVID-19 pandemic, during which febrile illnesses may have been misattributed to SARS-CoV-2, further complicating accurate diagnosis and surveillance of arboviral infections. While case location was based on residential address rather than place of infection, this georeferencing remains valuable for public health planning, as interventions are typically implemented at the community level.

Methodologically, the use of Kulldorff's space-time scan statistic introduces additional considerations. This approach assumes a discrete Poisson distribution and circular spatial scanning windows, which may not fully capture irregularly shaped clusters or adjust for potential confounders such as environmental or demographic variables. Cluster detection is also sensitive to the selection of scanning parameters, which may influence the size and significance of identified clusters. Additionally, the study's ecological design limits causal inference at the individual level, and interpretations of temporal trends and seasonality were based solely on descriptive visual analysis, without formal statistical modeling or seasonal decomposition. Finally, although the observed concentration of clusters in northeastern districts aligns with prior reports of higher vulnerability in these

areas, this was not a pre-specified hypothesis in our design and did not arise from our analytical model. We suggest that future studies explicitly investigate the structural and ecological factors behind this apparent spatial inequality.

Nonetheless, these findings provide a valuable overview of the situation of these diseases in Medellín and demonstrate the importance of individualizing their study, enhancing their understanding, and generating insights for their control. The study captured the emergence of dengue, chikungunya, and Zika in Medellín, highlighting the importance of understanding the spatio-temporal patterns of both emerging and endemic diseases. Furthermore, the study analyzed spatio-temporal patterns at a fine scale and neighborhood level, providing valuable insights for understanding disease patterns and effectively targeting interventions. Unlike broader regional studies, our results provide comuna-level evidence of co-circulation patterns and spatial heterogeneity within a single city, offering a finer scale of analysis.

While the present analysis does not include modeling of cluster determinants or forecasting techniques, it provides a high-resolution description of arboviral clustering patterns within a complex urban setting. For chikungunya and Zika, the clusters identified correspond to their respective introduction-driven epidemics and may not be reproducible in the absence of reintroduction. However, for dengue —which remains endemic in Medellín— we observed recurrent clusters over time, suggesting spatial persistence of transmission risk in certain areas. Although these results do not directly predict future outbreaks, they can guide targeted vector control in historically high-risk neighborhoods. Future studies should build on this work by incorporating explanatory variables and evaluating cluster stability across multiple epidemic cycles to improve prediction and prevention strategies.

In conclusion, this study provides a detailed characterization of the spatio-temporal dynamics and co-circulation patterns of dengue, chikungunya, and Zika in Medellín, Colombia, between 2013 and 2021. While the findings are not intended to be extrapolated beyond this specific urban context or to future epidemic scenarios, they offer relevant evidence for local decision-makers to guide vector control and surveillance efforts. The recurrent detection of high-risk clusters for dengue in northeastern districts suggests persistent vulnerabilities that require sustained attention. This localized analysis may also serve as a methodological reference for other hyperendemic cities seeking to better understand and respond to arboviral transmission at the intra-urban scale.

## Supporting information

**S1 Fig. Temporal distribution of the five dengue clusters detected in Medellín, Colombia (2013–2021), highlighting epidemic periods for each cluster over the full time series.**
(TIF)

**S2 Fig. Temporal distribution of the four Zika clusters detected in Medellín, Colombia (2015–2021), highlighting epidemic periods for each cluster over the full time series.**
(TIF)

**S3 Fig. Temporal distribution of the four chikungunya clusters detected in Medellín, Colombia (2014–2019), highlighting epidemic periods for each cluster over the full time series.**
(TIF)

**S4 Fig. Space-time clusters (A) and corresponding relative risk values (B) of dengue in Medellín, Colombia, from 2013 to 2015.** Note: The map was generated using municipality boundary basemaps from GADM (www.gadm.org) and open data from the Medellín city portal (https://www.medellin.gov.co/geomedellin/datosAbiertos/1043).
(TIF)

**S5 Fig. Space-time clusters (A) and corresponding relative risk values (B) of dengue in Medellín, Colombia, from 2013 to 2014.** Note: The map was generated using municipality boundary basemaps from GADM (www.gadm.org) and open data from the Medellín city portal (https://www.medellin.gov.co/geomedellin/datosAbiertos/1043).
(TIF)

**S6 Fig. Space-time clusters (A) and corresponding relative risk (B) of dengue in Medellín, Colombia, from 2017 to 2021.** Note: The map was generated using municipality boundary basemaps from GADM (www.gadm.org) and open data from the Medellín city portal (https://www.medellin.gov.co/geomedellin/datosAbiertos/1043).
(TIF)

## Acknowledgments

We extend our thanks to the Medellín Health Department for their thematic guidance and access to the data.

## Author contributions

**Conceptualization:** Jorge E. Salazar Flórez, Berta N. Restrepo, Laís Picinini Freitas, Mabel Carabali, Kate Zinszer.

**Data curation:** Jorge E. Salazar Flórez, Berta N. Restrepo.

**Formal analysis:** Jorge E. Salazar Flórez, Berta N. Restrepo.

**Funding acquisition:** Kate Zinszer.

**Investigation:** Berta N. Restrepo, Laís Picinini Freitas, Mabel Carabali, Kate Zinszer.

**Methodology:** Jorge E. Salazar Flórez, Berta N. Restrepo, Laís Picinini Freitas.

**Project administration:** Berta N. Restrepo, Laís Picinini Freitas, Mabel Carabali, Kate Zinszer.

**Resources:** Kate Zinszer.

**Supervision:** Laís Picinini Freitas, Mabel Carabali, Kate Zinszer.

**Validation:** Jorge E. Salazar Flórez, Berta N. Restrepo, Laís Picinini Freitas, Mabel Carabali, Gloria I. Jaramillo Ramírez, Cesar Garcia Balaguera, Brayan S. Avila Monsalve.

**Visualization:** Jorge E. Salazar Flórez, Berta N. Restrepo.

**Writing – original draft:** Jorge E. Salazar Flórez, Berta N. Restrepo.

**Writing – review & editing:** Jorge E. Salazar Flórez, Berta N. Restrepo, Laís Picinini Freitas, Mabel Carabali, Gloria I. Jaramillo Ramírez, Cesar Garcia Balaguera, Brayan S. Avila Monsalve, Kate Zinszer.

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
