## [Decision Letter · Decision Letter 0]

6 Mar 2025

Spatio-Temporal Analysis of the Distribution and Co-Circulation of Dengue, chikungunya, and Zika in Medellín, Colombia, from 2013 to 2021

Dear Dr. Salazar Flórez,

Thank you for submitting your manuscript to PLOS Neglected Tropical Diseases. After careful consideration, we feel that it has merit but does not fully meet PLOS Neglected Tropical Diseases's publication criteria as it currently stands. Therefore, we invite you to submit a revised version of the manuscript that addresses the points raised during the review process.

Please submit your revised manuscript within 60 days May 05 2025 11:59PM. If you will need more time than this to complete your revisions, please reply to this message or contact the journal office at plosntds@plos.org. Please include the following items when submitting your revised manuscript:

We look forward to receiving your revised manuscript.

Kind regards,

Jennifer K. Peterson

Academic Editor

Nigel Beebe

Section Editor

Shaden Kamhawi

co-Editor-in-Chief

Paul Brindley

co-Editor-in-Chief

**Journal Requirements:**

At this stage, the following Authors/Authors require contributions: Jorge Emilio Salazar Flórez, Berta N. Restrepo, Laís Picinini Freitas, Mabel Carabali, Gloria I. Jaramillo Ramírez, Cesar Garcia Balaguera, Brayan S. Avila Monsalve, and Kate Zinszer. Please ensure that the full contributions of each author are acknowledged in the "Add/Edit/Remove Authors" section of our submission form.

3) Some material included in your submission may be copyrighted. According to PLOSu2019s copyright policy, authors who use figures or other material (e.g., graphics, clipart, maps) from another author or copyright holder must demonstrate or obtain permission to publish this material under the Creative Commons Attribution 4.0 International (CC BY 4.0) License used by PLOS journals. Please closely review the details of PLOSu2019s copyright requirements here: PLOS Licenses and Copyright. If you need to request permissions from a copyright holder, you may use PLOS's Copyright Content Permission form.

Potential Copyright Issues:

i) Figures 2, 3, and (S7-S12). Please (a) provide a direct link to the base layer of the map (i.e., the country or region border shape) and ensure this is also included in the figure legend; and (b) provide a link to the terms of use / license information for the base layer image or shapefile. We cannot publish proprietary or copyrighted maps (e.g. Google Maps, Mapquest) and the terms of use for your map base layer must be compatible with our CC BY 4.0 license.

4) Thank you for stating "The data used in this study are publicly available. The raw and processed data used for the analyses are available at https://github.com/jemilios/Dengue." This link reaches a 404 error page. Please amend this to a new link or provide further details to locate the data.

2) State what role the funders took in the study. If the funders had no role in your study, please state: "The funders had no role in study design, data collection and analysis, decision to publish, or preparation of the manuscript.".

**Reviewers' Comments:**

Reviewer's Responses to Questions

**Key Review Criteria Required for Acceptance?**

**Methods**

-Are the objectives of the study clearly articulated with a clear testable hypothesis stated?

-Is the study design appropriate to address the stated objectives?

-Is the population clearly described and appropriate for the hypothesis being tested?

-Is the sample size sufficient to ensure adequate power to address the hypothesis being tested?

-Were correct statistical analysis used to support conclusions?

-Are there concerns about ethical or regulatory requirements being met?

Reviewer #1: The study does not present a clear testable hypothesis. It is clearly articulated with described goal, though.

Some of my concerns are:

According to description, is it correct to assume that only clinical cases were considered for the study? Please clarify

It is not well explained what is the nature of data being mapped (and provided by SIVIGILA): Is it the house of the cases?

In lines 188 -189, what was the purpose to transform cases into standard normal distribution? Visualization? What test was applied to check such distribution?

In addition, if data were z-transformed, I suppose it means that data follows a normal distribution, am I right? This is particularly important to clarify as the calculations for RR are assuming that data follows a normal distribution.

Line 194: What kind of discrete Poisson model was fitted? Generalized linear, Bayesian hierarchical? Mixed?

Explain further what "I()" means in the equation number 4. It is stated that it’s an indicator function, but it is an indicator of what? and why it is the same (1) for all windows? Multiply by 1 would mean that it makes no difference, so I don’t see the purpose of such value in the equation.

I have no ethical or regulatory concerns.

Reviewer #2: -Are the objectives of the study clearly articulated with a clear testable hypothesis stated?

Yes. However, similar studies that go further have already been published (e.g. Carabali, M. et al. 2020).

-Is the study design appropriate to address the stated objectives?

Partially. The methods are not clearly described, so the results are not reproducible.

-Is the population clearly described and appropriate for the hypothesis being tested?

No, it is not made clear from the outset that the data used correspond to potential cases and not proven cases, and the implications that this may entail.

-Is the sample size sufficient to ensure adequate power to address the hypothesis being tested?

Yes.

-Were correct statistical analysis used to support conclusions?

It is not easy to know since they do not adequately describe the methods used.

-Are there concerns about ethical or regulatory requirements being met?

No.

Reviewer #3: The statistical analyses are appropriately described and implemented. In the general comments section I mention some additional analyses that I think could improve the manuscript.

The github repository that is supposed to contain the data used in this analysis is missing.

**Results**

-Does the analysis presented match the analysis plan?

-Are the results clearly and completely presented?

-Are the figures (Tables, Images) of sufficient quality for clarity?

Reviewer #1: Results matched the analysis plan and are clearly presented. Some minor concerns are:

Figures S4, S5, and S6 which are supposedly showing spatio-temporal cluster are only showing temporal clusters (no spatial reference at all) and all of them corresponds to the same outbreak. So, they are no informative.

Reviewer #2: -Does the analysis presented match the analysis plan?

No completely. There are discussed results that cannot be inferred from the analyses. For example, the occurrence of waves of cases in dengue, Zika and chicungunya, the lack of seasonal patterns in Zika and chicungunya, and the seasonal patterns in dengue.

-Are the results clearly and completely presented?

No, it is not easy to follow the results, the figures are lightly mentioned in the text. There are many supplementary figures that are not properly described.

-Are the figures (Tables, Images) of sufficient quality for clarity?

No, there are repetitive figures, the quality is very poor, so it is not possible to read what is written. The labels are not appropriate and the northern array is incorrect.

Reviewer #3: Results match the analysis plan. The presentation of some of the results could be clearer. Details for how to improve them is provided in the general comments.

**Conclusions**

-Are the conclusions supported by the data presented?

-Are the limitations of analysis clearly described?

-Do the authors discuss how these data can be helpful to advance our understanding of the topic under study?

-Is public health relevance addressed?

Reviewer #1: Please take a look at "Summary and General Comments".

Reviewer #2: -Are the conclusions supported by the data presented?

No, the conclusions are too general and most of them were already known. There are no adequate comparisons with other studies already conducted. The author merely mentions other studies without putting them in context and discussing the implications related to his work.

-Are the limitations of analysis clearly described?

Partially. The limitations of the data are presented in the discussion section but not the limitations of the methods.

-Do the authors discuss how these data can be helpful to advance our understanding of the topic under study?

In a very general way, since they do not analyze the meaning of their results and what specific factors give rise to these clusters.

-Is public health relevance addressed?

Again, in a very general way. It is necessary to further understand the results, probably by carrying out other analyses that relate these clusters to socioeconomi, environmental, etc. factors.

Reviewer #3: The conclusions are in line with the results.

**Editorial and Data Presentation Modifications?**

Reviewer #1: The available captions for supporting figures should be more descriptive.

It is hard to elucidate the difference between neighborhood, district, and comunas. Please define them properly to ease the understanding of the spatial description of the clusters.

Reviewer #2: NA

Reviewer #3: See general comments below for a few suggestions of additional figures.

**Summary and General Comments**

Reviewer #1: Major comments:

The manuscript describes a study analyzing historical cases data and their spatial distribution. While these types of studies are locally relevant, I believe they may have limited appeal to a broader audience as it is limited only to the descriptive level.

Though it is stated in the Introduction that the aim is “to identify high-risk clusters for each disease individually and collectively to understand these arboviruses' co-circulation potential”, the manuscript doesn’t sufficiently describe the importance of achieving such goal. For instance, what is the significance of describing such patterns and how can these findings inform strategies for future application? In this sense, it seems to me that further analyses should be done to not limit the study to a narrow goal. In this way, the work can in addition be more innovative and original.

There is a longstanding issue partially resolved in recent years about this type of studies dealing with the place of infection’s acquisition. Though it is not properly explained in methods, I am assuming that authors are mapping the cases’ home addresses. If so, this study is assuming that infections are taking place at cases’ households. Last evidence has shown differential distribution of vectors between households and environments other than households as well as increased risk of infection in the latter. According to this, it is important to notice that clusters are not depicting infection risk or similar, but the spatial distribution of cases. A differential distribution of human population density within the city might potentially explain such distribution. In concordance, authors didn’t describe any density-based correction. What is the probability of the cluster of cases to be explained by a locally higher human population density? In my view, addressing these issues is crucial to align the study with current epidemiological and spatial knowledge.

Finally, identified clusters correspond to the same temporal outbreaks. So, in my understanding those are showing a one-time higher density of cases. I would expect for some cluster to be repeated in time and space, so preventive actions can be applied later. Since there are no spatial patterns repeated over time, it is hard to extract information that can be translated into something actionable. I wonder if a sensitive analysis was done to achieve highest performance of the spatial modeling.

Reviewer #2: Salazar Flórez et al.'s study on dengue, Zika and chicungunya in Medellín, Colombia, identified spatial patterns, with five dengue, fou Zika, and four of Chikungunya clusters concentrated in the northeast of the cit, as well as high-risk clusters with for all three diseases.

While the topic is crucial for priorizing intervention areas, the analysis primarily reiterates previously studies without providing novel insights. The study falls short in several aspects: temporal dynamics lack robust time series analysis for statistically significant seasons or trends; cluster detection methodology is insufficiently described, hindering reproducibility; spatial results are not thoroughly examined; and environmental and socioeconomic factors are overgeneralized without considering local conditions and interactions. These limitations suggest that the manuscript, in its current form, is unsuitable for publication. To enhance its relevance to academia and society, I suggest: i) conducting a more rigorous analysis that incorporates variables explaining the observed patterns, ii) considering local conditions and factor interactions in the analysis of environmental and socioeconomic influences, iii) Providing a more detailed description of the cluster detection methodology to ensure reproducibility, iv) performing a comprehensive time series analysis to support temporal dynamics conclusions, v) analyze the results in the context of previous research in the topic.

Please see comments below for more details:

*Major comments*

2. Introduction

- L116. Please provide an explanaition of why Medellín is a relevant city to conduct this analysis.

3. Methods

- L120-125. Please describe here that these cases include potential and proven cases.

- L193-194. Please mention and describe these univariate and multivariate techniques.

- L196. Please briefly describe the method and the assumptions it considers.

- L199. Which cylinder are the authors referring to?

- L211. Which specific window are the authors referring to?

- L217. Please explain what I() means.

- L217. Which program? What is the meaning of I() equal to 1?

- L224. Which location are the authors referring to?

- L229. Please describe how the maximum radius of the cylinder is considered.

- L229-232. Please explain how and why those values were defined.

4. Results

- L251. Please describe how the occurrence of waves and peaks was defined.

- L256. Please describe what is meant by the "apparent seasonal pattern". How can the authors conclude that if they did not perform any seasonal analysis. Also, with the y-axis scale in the figure is not possible to see values other than very high ones.

5. Discussion

- L340. Please explain why in western comunes if figs. 2 and 3 show the clusters in the Northeastern comunas.

- L341. How can the authors say that without a seasonal analysis?

- L343-344. Please explain what a "similar trend" means.

- L351. What do the authors mean by saying "this study showed a cyclical behavior until 2016"?. That is not reported in the results.

- L355. What do the authors mean by less rainfall? Less than what? Also, is Medellín a city with "less" rainfall?

- L356-358. Please explain which Medellín's trends? The results do not provide any analysis of trend behavior to say that the increase in Chicungunya and Zika in the country reflects the trend in Medellín.

- L359. Please explain wjat "discrepances" are being referring to.

- L361-362. Please explain what the authors mean by this statement.

- L372. Again, this type of statement cannot be concluded from this analysis. A trend analysis is needed to identify trends.

- L374. Wolbachia was not released in all comunas and not at the same time. Can this information help to understand the clusters?

- L383-389. Please describe the relevance of those other analyses performed at other sites to this analysis. Why are the values different? Any hypotheses?

- L391. This is the first time the "marginalized areas" are mentioned in this manuscript. The authors have not said anything about socioeconomical conditions so far to compare with national and local studies by others.

- L391-394. Why are those results different? Have conditions changed? Please contextualize those studies with yours and explain the source of the differences.

- L396-397. Why is this study relevant here? Please contextualize with your study.

- L400. It seems pretty obvious to me given that they are transmitted by the same mosquito. What may be interesting is to explaing why, according to the results of this analysis, there are clusters in which the three viruses do not co-circulate and cluster in which they do.

- L403-404. This is not a conclusion of this work.

- L408. Please explain the specific scores. There are different quantifications and the range of values are different between them.

- L409. Please explain what the reasons for the exception in Buenos Aires should be.

- L415. Please explain what the authors mean by "acceptance" in this context.

- L416-417. The role of socioeconomic and environmental factors in dengue transmission is widely known. The relevant information is which specific factors drive that phenomena and how (the mechanistic processes).

- L418-419. Please contextualize those results with your results.

- L424-425. Please explain the assumption or references saying that climatic variations in Medellin are similar. This city has a complex topography that may result in significant climate differences across the territory.

- L427-431. Please mention that the cited study considers data up to 2010. Conditions have likely changed in the last 15 years. Furthermore, explain the consequences and implications of those changes on your results.

- L431-433. Please explain why those results are relevant to your analysis.

- L442. Most cases are suspected cases, and only a small number of records are associated with an ELISA or PCR test.

- L444-447. Please explain why identifying clusters remains crucial to understanding the spatial distribution of cases resolves the limitation that the recorded address corresponds to the patient residence rather the place of infection.

- L449. Please describe the limitations of the methods used.

*Minor comments

1. Abstract

- L37. No need to mention the programming language.

- L41. Please mention that those cases correspond to potential cases.

2. Introduction

- L109-110. Prevalence of mosquito vectors is not an environmental factor. Environmental factors drive mosquitoes prevalence.

- L152-180. I do not consider those paragraphs relevant here. I suggest moving that information to an appendix or just writing the appropriate references.

3. Methods.

- L120. Please explain why this is an 'ecological study'.

- L133. What do the authors mean by LO and LN?

- L133-135. Please provide the appropriate reference.

- L140-146. I do not consider this is to be appropriate place for the ethics statement.

- L150. It is probably not "in Colombia" but for the INS. Please check.

- L150-152. Please provide the appropriate reference.

- L162. I think it means >=38.5 deg C.

- L188. Consider avoiding this type of sentence and going straight to the point.

- L201. Please refer to it as an equation instead of a formula and cross-reference it.

- L203. Please explain here how E is determined.

- Eq. 2 and 3 are not necessary. They are widely known.

- L236-237. In my opinion, this not relevant.

4. Results

- L250. Why are you referring to incidence if the figure shows cases.

- L276. How can people outside the country recognize the "Comunas". Please cross-reference the figures before starting to describe them. Also, in the figures is not possible to identify the text due to the poor quality.

5. Discussion

- L348-349. Please provide the appropriate reference.

- L363. This statement is quite obvious.

- L365. Explain that Antioquia is an administrative division with Medellín inside.

- L407-408. Please mention those "previous reports".

- L409. Please define what the authors mean by "district" here.

- L412. Please explain why the areas are important here.

*Figures*

- Figure 1. Panels A and B show repetitive information. Standarized cases are sufficient to recognize the times with the maximum number of cases.

- Figure 1B. y-label corresponds to the standarized number of cases, not number of cases.

- Figure 1. Please remove the days in the y-marks.

- Figure 2 and 3. Please avoid the "Notes". Describe what each panel means in the main legend text.

- Figure 2 and 3. Please correct the direction of the North arrow.

- Figure 2 and 3. Please add what the divisions mean.

- Figure 2 and 3. Please remove 'na' in the legend.

*Supplementary material*

- Consider reducing the number of supplementary figures to those that are really necessary.

- Please describe what is in each supplementary figure with the different panels.

Reviewer #3: This manuscript explores the spatiotemporal distribution of three arboviruses in Medellin, Colombia over the past decade. They provide evidence of co-circulation of all three pathogens and overlapping hotspots of dengue and Zika incidence shortly after the introduction of Zika in 2015-2016. The analysis is fairly limited and does not attempt to explain the timing of these clusters or the potential drivers of the observed spatial patterns. There are several issues that should be addressed to strengthen the results of this study.

Despite being described as a spatiotemporal analysis, there is no attempt to explain the observed patterns beyond the identification of several clusters of transmission and co-transmission. This analysis would be considerably strengthened if you were able to examine the association between these clusters and relevant socioeconomic and environmental factors at the neighborhood or district(communa) level.

Given the issues of underreporting and the potential for misdiagnosis among the three arboviruses with probable cases, it would be helpful to know what % of cases for each pathogen were confirmed versus probable. And if there is enough data, it could be useful to run the analysis on only confirmed cases to see if the patterns hold up.

There should also be some additional discussion of the potential for misdiagnosis and its implications on the observed spatiotemporal patterns. Have there been any seroprevalence studies for any of these arboviruses that would give an indication of the infection rate and how it compares to observed incidence?

I also wonder about the large increase in dengue cases that overlaps with the Zika outbreak. Could this be do to changes in surveillance during this time period. In most localities there were was increased surveillance and reporting of disease due to the risk of microcephaly and other Zika complications.

The analysis is of a single city in Colombia, but most of the Introduction is a very broad overview of the three pathogens and their recent epidemiology in Latin America. I think the Intro would benefit from at least a paragraph on what is known about the intra-city spatiotemporal patterns of these arboviruses and what the potential drivers of these patterns are in the region. As well as why this scale of analysis, as opposed to a broader regional or country-level analysis, is important.

It would be helpful to have a map of the mean annual incidence rates at the neighborhood level for each pathogen. Since the existing maps don’t show RR outside the clusters it is hard to visualize the overall pattern of incidence in the city.

Minor comments:

Line 65 of Author summary: “This analysis addresses a critical gap in understanding the intra-urban spread of arboviruses in tropical cities”. I don’t see how this analysis addresses intra-urban spread as there is no analysis or discussion of pathogen spread from one location to another.

Lines 96-97: “For instance, …” – The use of this phrase is misplaced as this sentence is unrelated to the previous sentence.

Lines 221-222: Why do c and p represent the number of cases and population size in each district (communa) and not at the neighborhood-level as I thought that was the level of analysis being conducted?

Line 229: Can you explain further how considered the maximum radius of the cylinder? I didn’t see any mention in the Results to an examination of different values for the cylinder radius.

Line 272 – Dengue spatial and temporal distribution: I think it needs to be made a bit clearer that the main result of 5 dengue clusters is when the scan is done for entire time period. Initially I was confused about the difference between this result and the mention of differing number of clusters for the shorter time periods mentioned at the end of this section. It might also be useful to state either here or in the Methods why these specific additional time periods were chosen.

Line 285: Why does this sentence start with “Conversely”? It does not seem to contradict the findings in the previous section.

Lines 323-324: You give the length of clusters 3 and 5, but not the other three.

Figure S12 – The RR displayed in this figure doesn’t seem to match the RR values in Fig 2D. Is that due to the differing time periods in the two maps?

Line 337-338: It is not clear what this sentence means or how the identification of dengue clusters is evidence of cyclical viral activity.

Line 351: It would be useful to also mention the behavior observed post-2016.

Line 357: CHIKV wasn’t present before 2014, so I wouldn’t call the observed pattern “increased incidence”, especially since incidence declined strongly by the end of 2017.

Line 359: Discrepancies in what?

Lines 371-372: Please edit this sentence for grammar and clarity.

Line 378: I assume you are referring to the COVID-19 pandemic?

Lines 392-394: Can you provide some explanation for why prior studies found differing patterns to the current study?

Lines 424-425: What findings remain significant? You didn’t look at climate variability in your analysis.

PLOS authors have the option to publish the peer review history of their article (what does this mean? ). If published, this will include your full peer review and any attached files.

**Do you want your identity to be public for this peer review?** For information about this choice, including consent withdrawal, please see our Privacy Policy .

Reviewer #1: No

Reviewer #2: No

Reviewer #3: No

**Figure resubmission:**

**Reproducibility:**



---

## [Decision Letter · Decision Letter 1]

25 Jun 2025

Spatio-Temporal Analysis of the Distribution and Co-Circulation of Dengue, chikungunya, and Zika in Medellín, Colombia, from 2013 to 2021

Dear Dr. Salazar Flórez,

Thank you for submitting your manuscript to PLOS Neglected Tropical Diseases. After careful consideration, we feel that it has merit but does not fully meet PLOS Neglected Tropical Diseases's publication criteria as it currently stands. Therefore, we invite you to submit a revised version of the manuscript that addresses the points raised during the review process, specifically the comments of reviewer 1.

Please submit your revised manuscript within 60 days Jul 25 2025 11:59PM. If you will need more time than this to complete your revisions, please reply to this message or contact the journal office at plosntds@plos.org. Please include the following items when submitting your revised manuscript:

We look forward to receiving your revised manuscript.

Kind regards,

Jennifer K. Peterson

Academic Editor

Nigel Beebe

Section Editor

Shaden Kamhawi

co-Editor-in-Chief

Paul Brindley

co-Editor-in-Chief

**Additional Editor Comments:**

Please address the comments of reviewer 1.

**Reviewers' Comments:**

Reviewer's Responses to Questions

**Key Review Criteria Required for Acceptance?**

**Methods**

-Are the objectives of the study clearly articulated with a clear testable hypothesis stated?

-Is the study design appropriate to address the stated objectives?

-Is the population clearly described and appropriate for the hypothesis being tested?

-Is the sample size sufficient to ensure adequate power to address the hypothesis being tested?

-Were correct statistical analysis used to support conclusions?

-Are there concerns about ethical or regulatory requirements being met?

Reviewer #1: Methods are better explained in this version.

Related to hypothesis: It is not clear why authors hypothesized that socioeconomically disadvantaged conditions presented specifically at the northeastern districts render the areas more susceptible to spatio-temporal cluster. Is it a results-driven hypothesis?

Though the manuscript improved, I believe that the fundamental concerns of the methodology were not appropriately resolved:

The work does not make analysis aimed to understand the nature of the clusters. So, instead of providing information leading to potential improvement of control activities, it presents an analytical narrative of historical reports.

Authors claimed that the purpose of these analyses is to “enhance future outbreak predictions, enabling targeted resource allocation for prevention and control efforts”. All clusters that authors found were tied to a specific epidemic. I wonder how reproducible the found clusters are if only one epidemic was analyzed. If similar clusters would be found through multiple epidemics, it could be stated that reproducible pattern in time is happening and enhanced predictions could arise. For example, the found clusters for Zika and Chikungunya are temporally located within epidemics that happened due to introductions into a naïve population. There haven’t been more epidemics since then as the populations is no longer naïve. How these results improve control of these diseases?

Reviewer #3: (No Response)

**Results**

-Does the analysis presented match the analysis plan?

-Are the results clearly and completely presented?

-Are the figures (Tables, Images) of sufficient quality for clarity?

Reviewer #1: Results matched the analysis plan. Besides the big improvement of the manuscript, I think the problem is the plan itself. As it's mentioned in the previous review: "While these type of studies are locally relevant, I believe they

may have limited appeal to a broader audience as it is limited only to the descriptive level". There's room for a lot of improvements to the work.

Reviewer #3: (No Response)

**Conclusions**

-Are the conclusions supported by the data presented?

-Are the limitations of analysis clearly described?

-Do the authors discuss how these data can be helpful to advance our understanding of the topic under study?

-Is public health relevance addressed?

Reviewer #1: Conclusiones are supported by data and results.

Some limitations are clearly described.

My biggest concerns are related to public health relevance: the study is highly local and results are not extrapolable to other settings or times.

Reviewer #3: (No Response)

**Editorial and Data Presentation Modifications?**

Reviewer #1: Line 103: Improve references and data of asymptomatic infections.

Line 221: A reference should be used when mentioned SaTScan for first time. Also, the version of the software used should be reported.

The sentence “The city's administration divides Medellín's territory into 21 units known as 'Comunas'.” Is repeated

Improve position/size of legends as they overlap maps.

Reviewer #3: (No Response)

**Summary and General Comments**

Reviewer #1: The manuscript is improved in regards to basic elements exposed by reviewers. But in my point of view, fundamental improvements should be done, i.e. take the data to broader analysis and results, so the analysis can acquire bigger relevance and can potentially be generalizable to other settings.

Reviewer #3: All of my comments and suggestions from the previous version of this manuscript have been addressed.

PLOS authors have the option to publish the peer review history of their article (what does this mean? ). If published, this will include your full peer review and any attached files.

**Do you want your identity to be public for this peer review?** For information about this choice, including consent withdrawal, please see our Privacy Policy .

Reviewer #1: No

Reviewer #3: No

**Figure resubmission:**

**Reproducibility:**



---

## [Editor Report · Decision Letter 2]

10 Aug 2025

Dear Professor Salazar Flórez,

We are pleased to inform you that your manuscript 'Spatio-Temporal Analysis of the Distribution and Co-Circulation of Dengue, chikungunya, and Zika in Medellín, Colombia, from 2013 to 2021' has been provisionally accepted for publication in PLOS Neglected Tropical Diseases.

Best regards,

Jennifer K. Peterson

Academic Editor

Nigel Beebe

Section Editor

Shaden Kamhawi

co-Editor-in-Chief

Paul Brindley

co-Editor-in-Chief

Line 61 is a sentence fragment (missing article and verb)- please resolve.

---

## [Editor Report · Acceptance letter]

Dear Professor Salazar Flórez,

We are delighted to inform you that your manuscript, "Spatio-Temporal Analysis of the Distribution and Co-Circulation of Dengue, chikungunya, and Zika in Medellín, Colombia, from 2013 to 2021," has been formally accepted for publication in PLOS Neglected Tropical Diseases.

Best regards,

Shaden Kamhawi

co-Editor-in-Chief

Paul Brindley

co-Editor-in-Chief
